# Factors influencing birth preparedness and complication readiness among childbearing age women in Thatta district, Sindh

Ruquia Noor[1], Farhana Shahid[2], Muhammad Zafar Iqbal Hydrie[3]*, Muhammad Imran[4], Syed Hassan Bin Usman Shah[5]

**1** Department of Respiratory and Critical Care Technology, Dow Institute of Medical Technology, Karachi, Pakistan, **2** Faculty of Life Sciences, Shaheed Zulfikar Ali Bhutto Institute of Science and Technology, Dubai, United Arab Emirates, **3** School of Public Health, Dow University of Health Sciences, Karachi, Pakistan, **4** Institute of Medical Technology, JSMU, Karachi, Pakistan, **5** The Kirby Institute, UNSW, Sydney, Australia

* zafar.hydrie@duhs.edu.pk, zafarhydrie@gmail.com

## Abstract

### Introduction

Birth preparedness and complication readiness (BPCR) is a broad system to increase the practice of trained health provision at the time of childbirth and the key interventions to decrease mothers' and newborns' death. However, its status and influencing factors have not been well studied at different levels in the study area. The current study aimed to assess the BPCR status and explore its associated factors influencing BPCR among childbearing age women in Thatta, District of Sindh.

### Methods

This community-based cross-sectional study was conducted among 770 recently delivered mothers from October 2016 –September 2017, recruited using a multistage cluster sampling technique. A structured validated close-ended questionnaire measuring BPCR knowledge and practices was used for the interviews. The results were analyzed by means of the Chi-square test, and a binary logistic regression model was used to determine the factors influencing BPCR.

### Results

The overall response rate was 94.6%, with a low BPCR status. Out of 770 participants, only 163 (21.2%) were well prepared, while 607 (78.8%) were not prepared for safe childbirth and its complications. A small proportion of women knew about the serious warning signs of pregnancy, labour, childbirth and the postpartum period (16.2%), (15.3%) and (22.7%) respectively. Antenatal care (ANC) checkup ($P < 0.001$), cost of ANC checkup ($p = 0.016$), place of birth ($p = 0.014$), awareness of serious warning signs during pregnancy ($p = 0.001$) and awareness of serious warning signs during the postpartum period ($p < 0.001$) were found to be significant predictors of BPCR.

**Data Availability Statement:** Data cannot be shared publicly because of university policy. Data are available from the JSMU Institutional Data Access / Ethics Committee (contact via the Director of Institute of Medical Technology at email:

IMT@jsmu.edu.pk) for researchers who meet the criteria for access to confidential data.

**Funding:** The author(s) received no specific funding for this work.

**Competing interests:** The authors have declared that no competing interests exist.

## Conclusion

The proportion of women who were well prepared for birth and its complications was low. It is recommended to organize community-based education campaigns and improve the quality of MNCH services at every level to increase BPCR among women in Sindh.

## Introduction

Maternal mortality is a major public health issue, with over 295,000 deaths in women due to pregnancy and childbirth related problems in 2017. Unfortunately, 94% of these maternal deaths occur in developing countries, according to the World Health Organization (WHO) [1]. Pregnancy may lead to sudden, unpredictable, dangerous complications, leading to increased mortality and lifelong morbidity for the mother and her baby. The magnitude of the problem is quite severe with approximately 810 women dying from preventable causes related to unsafe pregnancy and child delivery every day in 2017 [2].

One of the primary goals of sustainable development goals (SDGs) is reducing maternal mortality [2]. The key to reduce the maternal mortality ratio and to improve maternal and neonatal health is to increase the availability of skilled health attendants during pregnancy and child delivery [3]. Birth preparedness and complication readiness (BPCR) is the procedure of arranging in advance for safe and normal childbirth and the necessary actions required to be taken in case of any emergency during childbirth [4]. BPCR facilitates safe delivery by planning before birth. It is measured by using a series of questions, including awareness of at least eight 'serious warning signs' during pregnancy, labour & childbirth and the postpartum period. Additionally, planning for a trained birth attendant, saving money for pregnancy, considering healthcare facility for delivery, planning for transportation, and identifying blood donors in case of emergency [5]. If women identified four or more components out of those mentioned above, they were considered prepared for BPCR. This scoring method has been previously used to assess the status of birth preparedness and complication readiness [6].

BPCR plan reduces delays in deciding to seek care in two ways. Firstly, it encourages pregnant women to plan ahead to have trained health care providers at every birth. If women and their families plan to seek care before the onset of labour and successfully follow this plan throughout childbirth, then the women will definitely reach the health care facility before any type of complication becomes critical. Secondly, a complication readiness plan raises awareness of serious warning signs among women, families and communities, thereby improving problem recognition and reducing the delay in deciding to seek care [7]. Literature suggests that around 35% of the overall disease burden affecting the population is associated with pregnancy and childbirth-related conditions in Pakistan [8].

BPCR matrix was introduced by the Maternal and Neonatal Health (MNH) Program of JHPIEGO; an international non-profit organization. This matrix aims to address the three delay model parameters, which are three types of delays at various levels. It includes delays in seeking care, reaching care, and receiving adequate care once at the point of service to the pregnant woman.This matrix involves her family, her community, health care providers, health institution, and policy-makers during pregnancy, childbirth, and the postpartum period. Thus, BPCR is a key strategy in the safe motherhood program [9].

Maternal deaths are more prevalent in developing countries due to a lack of BPCR [10]. A cross-sectional survey conducted among Kenyan and Tanzanian women of reproductive age reported that only 11.4% and 7.6% of women were well-prepared for birth and its

complications [11]. In the Upper East Region of Ghana, an analytical cross-sectional study conducted among mothers to assess the prevalence of BPCR found that less than 15% mothers were prepared [12].

In Riyadh, a community based survey reported that 21.1% of the mothers knew about swollen hands or faces during labour while 23.1% had knowledge about prolonged labour (> 12 h). Moreover, only 26.3% of respondents knew about foul-smelling vaginal discharge as a danger sign [13]. Thus, it is necessary to improve preventive behavior and increase knowledge of pregnant women about the danger signs of pregnancy and childbirth by promoting birth and emergency planning in advance [14]. Maternal mortality is a serious public health issue in Pakistan, and according to the 2019 Maternal Mortality Survey, it is estimated that the maternal mortality ratio (MMR) in Pakistan is 186 deaths per 100,000 live births for the 3-years [15]. The MMR is 26% higher in rural areas than in urban areas. The survey found that the MMR in Balochistan (298), Punjab (157), Sindh (224) and Khyber Pakhtunkhwa (165) per 100,000 live births [15]. Unfortunately, this ratio is much higher than the South Asian figures of 157 per 100,000 live births [16]. The contraceptive prevalence rate (CPR) in rural Sindh was reported to be as low as 21.4 percent, with only 41% of pregnant women receiving four or more antenatal care (ANC) visits and only 58.2 percent of deliveries taking place in a health facility [17].

In Pakistan, recent literature showed increased maternal and perinatal mortality & morbidity among women who have not had an antenatal check-up. Haemorrhage, anemia, preeclampsia/eclampsia, dystocia, and sepsis are the most common causes [18]. A cross-sectional study conducted in Wah Cantt, Pakistan revealed that only 34.1 percent of women were prepared for child delivery and its emergency complications. The study further found that, despite having access to free medical care and education, women were unaware of birth preparation [19].

Community based programmes help to promote maternal health by raising awareness of danger signs of pregnancy and child delivery preparation in advance to overcome the complications during pregnancy. A cluster randomized controlled trial was conducted in Matiari and Hyderabad, Sindh Province which showed that community engagement programs for male and female stakeholders increased some measures of knowledge regarding complications of pre-eclampsia in low-resource settings [20].

To our best knowledge, there is limited literature available investigating the factors that influence BPCR in Pakistan, especially in rural areas of Sindh. Such studies are essential to understand the underlying factors, increase awareness and practice of safe childbirth, and advocate for health reforms. Therefore, this study aimed to explore the factors influencing BPCR among recently delivered mothers in rural areas such as Thatta district in Sindh. Hopefully, this study will provide important information regarding the factors and status of BPCR in rural Pakistan. They will also help the Ministries of health and social organization, policy planners and public health specialists to identify and construct community-based interventions which will encourage pregnant women and their families to be more prepared for BPCR and overcome the first delay.

## Methodology

A community-based cross-sectional study was conducted among (15–49 years old) recently delivered mothers in Thatta, district, Sindh from October 2016 to September 2017. A multistage cluster sampling was used for the selection of study participants (Fig 1). In the first stage, lists of houses were collected. Thatta is divided into Four Talukas (Sections) and 30 Union council, and each union council has around 18–20 village (deh). Out of a total of four Talukas, two Talukas were chosen by using simple random sampling. In every Taluka, three union

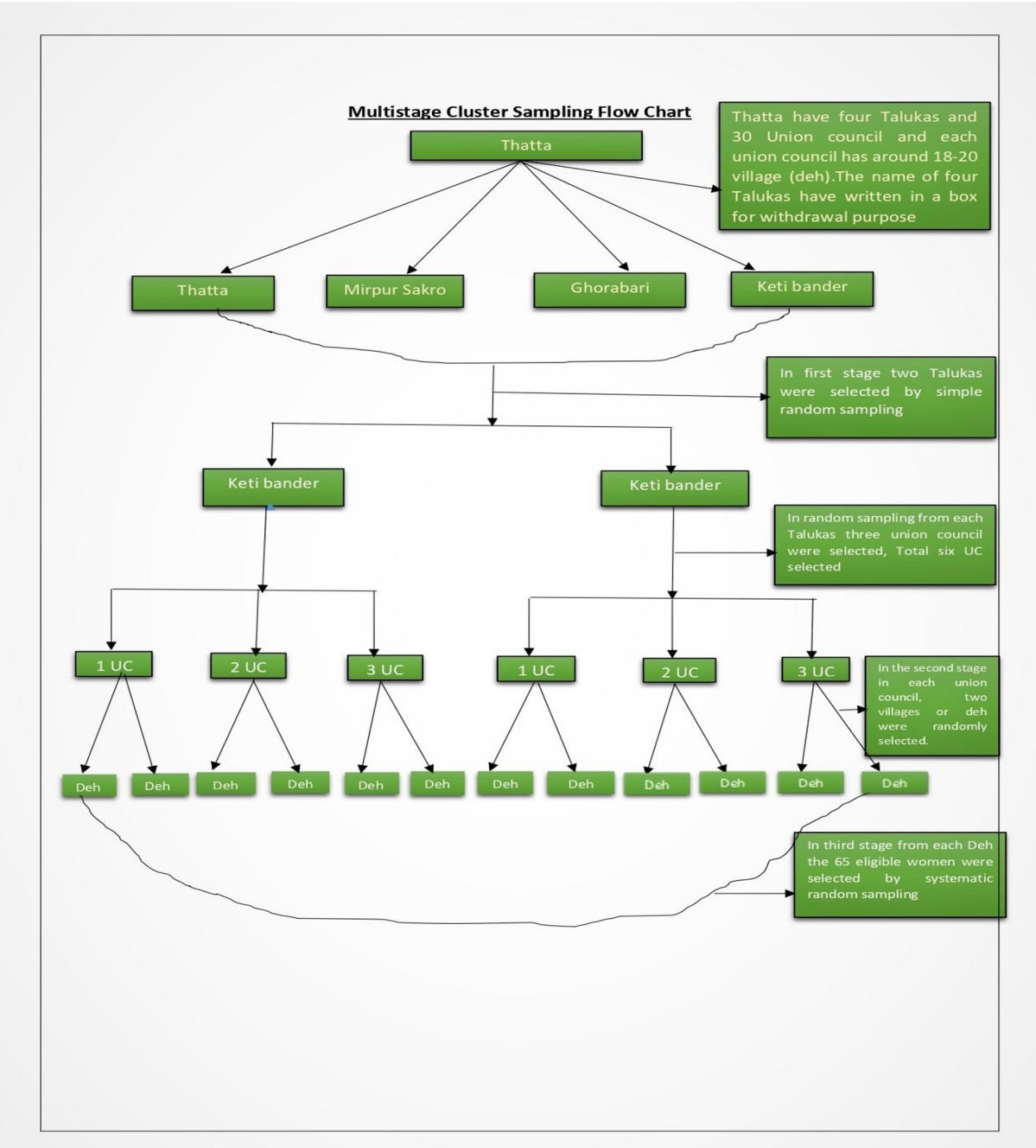

**Fig 1. Sampling technique.**

councils were randomly selected. In the second stage in each union council, two villages or deh were randomly selected. In the third stage, a household of the selected village where a woman had delivered their baby during the last 12 month was selected for interview. In one village, 65 women from households were selected by systematic random sampling. Verbal consent was taken from the participants before data collection and noted on the forms. The survey was conducted after ethical approval by the IRB of Jinnah Sindh Medical University (Reference

No: JSMU/IRB/2015/-17) dated 19th November 2015. For cultural reasons, trained female team-members conducted interviews of the participants. Subjects were informed that they could withdraw from participating at any time during the study.

Sample size was determined by using the "Open Epi" software (Sample Size for Frequency in a Population) by considering the proportions based on the following assumptions. Since there were no previous data regarding BPCR available in nearby rural districts, the percentage of women practicing BPCR in rural Ethiopia was taken, which was 37%. Based on the above assumptions, and for a 95% CI with a 5% margin of error, the calculated sample size was 717. An additional 10% allowance for the non-response rate was added. Since multistage cluster sampling was used, a design effect of two was considered as a correction for this sampling technique. The final sample size was 770 mothers [21]. Pregnant women and those who were mentally or physically challenged were excluded from the study. The structured questionnaires were adapted from a survey tool developed by JHPIEGO Maternal, Neonatal Health Program [22]. As part of the data collection procedures, the questionnaire consisted of five parts, the first part of the questionnaire assessed the demographic characteristic of the participants, such as age, income, profession, education and ethnicity, the second part assessed the variables regarding antenatal check-up such as number of ANC visit and cost of ANC check-up, third part assessed the variables regarding obstetric factors such as the choices for the place of delivery during previous pregnancy while the fourth part assessed the variables of knowledge of specific danger signs as showed in Table 1. The last part assessed the BPCR status that identified place of delivery, plan for skilled attendant at birth, plan for transport during emergency, saving money for obstetric emergency and preparing blood donor. Composite score and mean computed score for those who answered yes for 4 or more components out of six components of BPCR are defined as "well prepared" while those who answered yes for less than 4 components out of 6 were defined as "Less prepared" for BPCR. All data were entered in the IBM SPSS Statistics for Windows, version 23.0 (IBM Corp., Armonk, NY, USA). Continuous variables were summarized by reporting mean & standard deviation (SD), and categorical

**Table 1. Knowledge of key danger signs among childbearing age women in Thatta district Sindh, 2017 (n = 770).**

| Key danger signs during pregnancy | n (%) |
| --- | --- |
| Severe vaginal bleeding | 156(20.3) |
| Swollen hands and face | 24(3.1) |
| Blurred vision | 7(0.9) |
| All of the above | 126(16.4) |
| None | 457(59.4) |
| **Key danger signs during Labour and childbirth** | |
| Severe vaginal bleeding | 134(17.4) |
| Prolonged labour (> 12 hours) | 20(2.6) |
| Convulsions | 6(0.8) |
| Retained placenta | 13(1.7) |
| All of the above | 118(15.3) |
| None | 479(62.2) |
| **Key danger signs during postpartum** | |
| Severe vaginal bleeding | 216(28.1) |
| Foul lochia | 4(0.5) |
| High fever | 24(3.1) |
| All of the above | 175(22.7) |
| None | 351(45.6) |

variables were expressed in frequency tables, percentages and pie or bar chart. The chi-square test analyzed the association between variables. Following unadjusted analyses, multivariate logistic regression evaluated factors that influence BPCR. The statistical significance among variables was considered at a p-value < 0.05.

## Results

### Socio-demographic characteristics

For the purpose of the study, a total of 815 participants were approached, out of which 770 completed the questionnaire giving a 94.6% response rate. The mean age of the participants was 28.58 (SD±4.96) years, and more than half n = 498 (64.7%), were between the age group of 20–30 years. The majority of them were Muslims, n = 717 (93.1%), while Sindhi was the predominant n = 649 (84.3%) local language. By profession, the majority, n = 646 (83.9%) of the participants were housewives with no independent income source, while n = 124 (16.1%) women were earners. About three-quarter n = 503 (70.5%) participants had no formal education taken as uneducated, while only n = 9 (1.2%) had a university education. More than 65% (n = 521) husbands, were also uneducated.

Fig 2 showed that only half of the deliveries n = 417 (54.2%) occurred at a health facility, whereas nearly half n = 353 (45.8%) women delivered their last baby at home.

**Knowledge of key obstetric danger signs.**   The knowledge of key danger signs among childbearing age women in Thatta district Sindh is shown in Table 1 below.

**Knowledge of participants about danger signs during pregnancy.**   A relatively small proportion 156 (20.3%), 24 (3.1%) and 7 (0.9%) of the participantsspontaneously mentioned severe vaginal bleeding, swollen hands and face and blurred vision as danger signs during

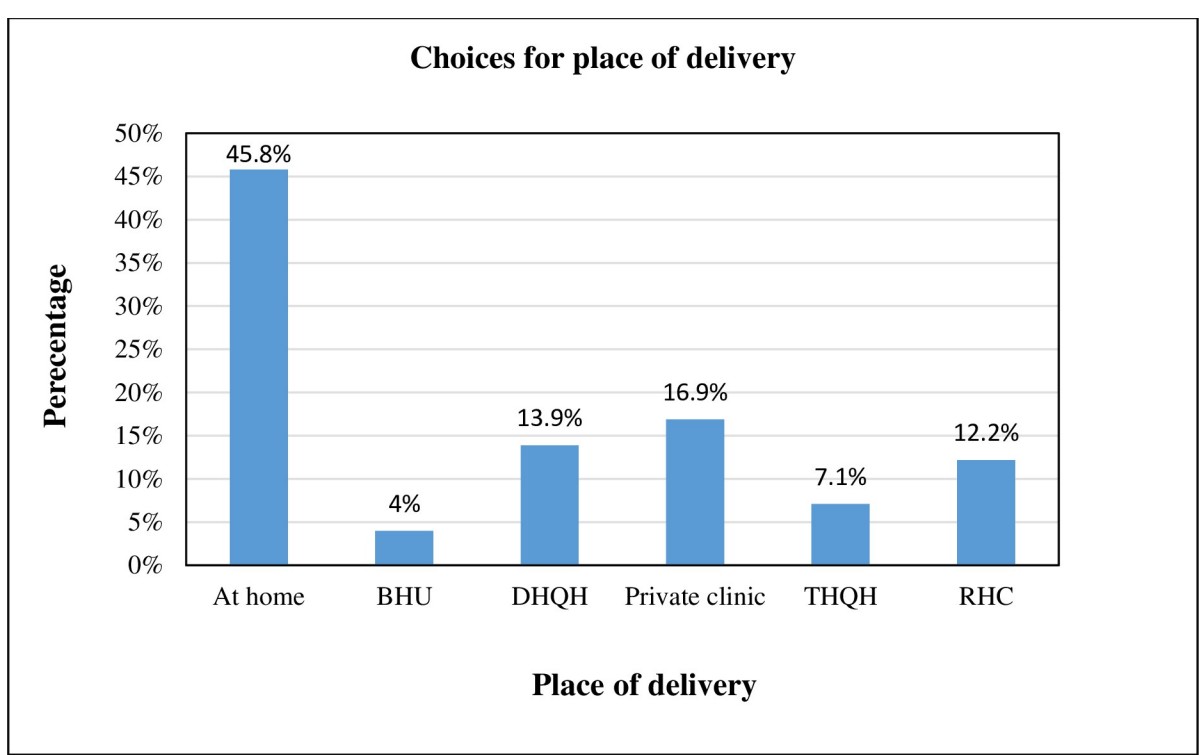

**Fig 2.  Preferences of the study participants for birth settings.**

**Table 2. BPCR practice among childbearing age women in Thatta district, Sindh, 2017 (n = 770).**

| Variable | n (%) |
|---|---|
| Identify the place of delivery | 371(48.2) |
| Saved money | 364 (47.3) |
| Identify transport for delivery | 314 (40.8) |
| Identify a skilled birth attendant | 307 (39.9) |
| Identify blood donor | 140 (18.2) |
| Awareness of at least eight key serious warning signs during pregnancy, labour & childbirth and the postpartum period | 53 (6.9) |

pregnancy, respectively. Around n = 457 (59.4%) participants did not mention any danger signs, n = 187 (24.3%) mentioned at least one key danger sign, while n = 126 (16.4%) mentioned all three key danger signs.

**Knowledge of participants about danger signs during labour and childbirth.** One hundred and thirty-four (17.4%), 20 (2.6%), 6 (0.8), and 13 (1.7%) of the participants mentioned severe vaginal bleeding, prolonged labour (>12 hours), convulsions and retained placenta as danger signs during labour and childbirth, respectively. Four hundred and seventy-nine (62.2%) participants did not mention any danger signs, 173 (22.5%) mentioned at least one key danger sign, while 118 (15.3%) mentioned all three danger signs of labour and childbirth.

**Knowledge of participants about danger signs during postpartum.** Two hundred and sixteen (28.1%), 4 (0.5%) and 24 (3.1%) of the participants spontaneously mentioned severe vaginal bleeding, foul lochia and high fever as danger signs during the postpartum period, respectively. Around n = 351 (45.6%) of the participants did not mention any danger signs, only n = 244 (31.7%) participants mentioned at least one key danger sign, and n = 175 (22.7%) mentioned all three danger signs during postpartum.

The status of BPCR among recently delivered mothers in this rural district was found to be n = 163 (21.2%). The majority n = 371 (48.2%) of the participants reported that they identified place of delivery for the birth of their baby. In comparison, less than half n = 364 (47.3%) had saved money for their childbirth, n = 314 (40.8%) and n = 307 (39.9%) had identified transport and skilled birth attendant for delivery, and n = 140 (18.2%) of the participants identified blood donor who would donate blood in case of an obstetric emergency. However, only 49 (6.4%) had awareness about serious warning signs of (pregnancy, labour & childbirth and the postpartum period) as shown in Table 2.

## Score on six basic elements of BPCR

Among the 770 participants, over 163 (21.2%) scored at least four out of six components of BPCR. They were considered as 'well prepared' for birth in terms of choice of health facility to deliver, preparation for transportation, blood donors in case of emergencies, identification of skilled birth attendants, knowledge of danger signs, and saving money for expenses of their delivery in their last pregnancy. In contrast, the remaining n = 607 (78.8%) scored less than four and were considered 'not/less prepared'. Other BPCR components can be seen in Fig 3.

Factors found to be associated with BPCR in the bivariate analysis were then entered into a multivariate analysis model for further analysis. According to the Backward Likelihood-Ratio multivariate logistic regression analysis, ANC visits (p<0.001), cost of ANC checkup

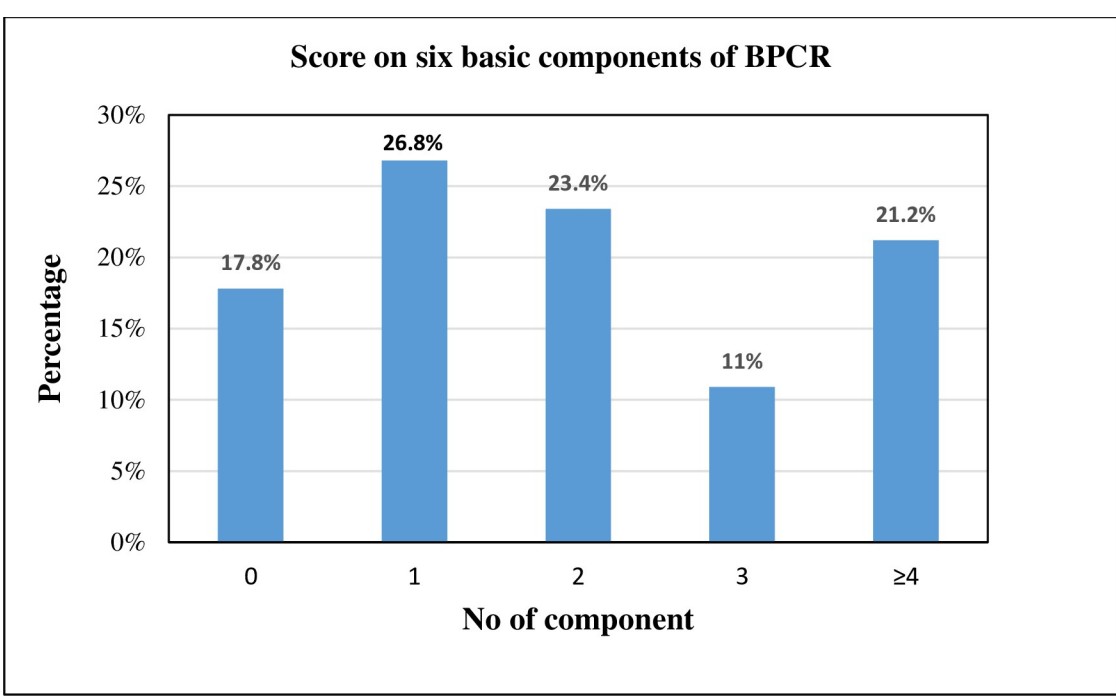

**Fig 3. Trends in basic BPCR component scores among study participants, n = 770.**

(p = 0.016), place of birth (p = 0.014), awareness of serious warning signs during pregnancy (p = 0.001) and awareness of serious warning signs during postpartum (p< 0.001) were found to be significant predictors of BPCR as show in in Table 3.

Those women who had attended ANC checkup four or more times during their last pregnancy were positively associated with BPCR [Adjusted Odds ratio (aOR) = 3.78, 95% CI: 2.53, 5.64] than those mothers who attended ANC checkup less than four times. Furthermore, those women who had thought ANC checkup was not costly showed a significant association with BPCR [aOR = 1.67, 95% CI: 1.10, 2.55] compared to those who thought ANC checkup was costly.

In addition, awareness of serious warning signs during pregnancy and postpartum were significantly associated with BPCR. Those women who were not knowledgeable about serious warning signs during pregnancy were less likely to be prepared for birth and its complication than those who were knowledgeable [aOR = 0.42, 95% CI: 0.25, 0.71].

Those women who were not knowledgeable about serious warning signs during the postpartum period were less likely prepared for BPCR [aOR = .44, 95% CI: .28, .68] compared to those who were knowledgeable. Additionally, mothers who delivered their last baby at health institution were two times more likely to be prepared [aOR = 2.14, 95% CI: 1.16, 3.94] for their birth and its complication compared to those who delivered at home (Table 3).

## Discussion

The purpose of this study was to assess the status of birth preparedness and explore the factors that influence BPCR in a rural district of Sindh. The practice of birth preparedness and complication readiness in this study was found to be low i.e. 21.2%, as compared to other developing countries studies such as Ethiopia (30.2%) and India (48.8%) [23,24], while still much lower than medical facility based studies as in West Bangal (75%), Ghana (78%) and Nigeria

**Table 3. Multivariate analysis of factors that influence BPCR (n = 770).**

| Variables | | | 95% C/I for OR | |
|---|---|---|---|---|
| | Significance | aOR | Upper | Lower |
| **Place of delivery** | | | | |
| At home | 0.000 | | | |
| BHU | .014 | 2.145 | 1.168 | 3.940 |
| DHQH | .151 | .505 | .199 | 1.282 |
| Private clinic | .138 | .595 | .300 | 1.181 |
| THQH | .135 | .615 | .325 | 1.164 |
| RHC | .191 | 1.895 | .728 | 4.936 |
| **ANC visits** | | | | |
| 0 | 0.000 | | | |
| <4 | .998 | .300 | 0.000 | |
| ≥4 | 0.000 | 3.783 | 2.533 | 5.469 |
| **Do you think checkup was costly** | | | | |
| Yes | 0.016 | | | |
| No | 0.016 | 1.679 | 1.103 | 2.555 |
| **Awareness of serious warning signs during pregnancy** | | | | |
| Awareness | | | | |
| No awareness | 0.001 | .426 | .256 | .710 |
| **Awareness of serious warning signs during postpartum** | | | | |
| Awareness | | | | |
| No awareness | 0.000 | .445 | .288 | .687 |

(72.6%) [25–28]. This comparable low proportion of birth preparedness and complication readiness in rural areas might be due to low education levels, shortage of health services and poor or inadequate guidance about BPCR in antenatal checkup during pregnancy.

In contrast, similar findings were seen in Rwanda (22.3%) and Ethiopia (22.2%) [29,30]. Such findings were also found in a local study conducted in Pakistan (23.6%) [31]. Even a lower level of BPCR was seen in other studies conducted in developing countries such as Tanzania (7.6%), Nepal (7.3%), and Bangladesh (12%) [12,14,32]. This variation might be due to socio-cultural and study settings differences as well as the implementation of related health programs.

In this study, the overall level of awareness regarding serious warning signs during pregnancy, labour and postpartum period was poor. The study revealed that only 16.4%, 15.3% and 22.7% of the women could at least identify serious warning signs associated with pregnancy, childbirth/ labour and the postpartum periodrespectively. The findings of serious warning signs during pregnancy, childbirth/labour, and postpartum in this study were much lower than seen in the study conducted in Bangladesh 61.4%, 61.5% and 40.5% during pregnancy, childbirth/ labour and postpartum period, respectively, which indicates a decreased chance of pregnancy complications or poor outcome [33]. These variations in different studies may be due to the level of awareness developed by the lady health worker (LHW) and community mid wives (CMWs) or due to the number of ANC visits and information given by the health care provider [34].

However, the knowledge of serious warning signs during pregnancy, childbirth/labour and postpartum was higher than in Bangladesh, which was only 5% and 6% during pregnancy and delivery [32]. While in rural Rwanda, 6.6% could mention three or more key danger signs during all three periods and in India 18%, 0%, and 4% during pregnancy, childbirth and postpartum period respectively [35,36].

In this study, of the four components of BPCR identified place of delivery (48.2%) was the most frequently adopted component. Other components included were saved money for childbirth (47.3%), identified skilled birth attendant (39.9%) and identified mode of transportation (40.8%). The BPCR component findings of this study are similar to that seen in the Nigerian study, which identified the mode of transport in 23.5%, while a skilled birth attendant in 45.7%, saved money in 42.4% and place for delivery in 55.7%, respectively [34].

In contrast to this study, the findings were lower in the regional Bangladesh study, where only a minority of participants had the level of preparedness with 12% identifying the place for delivery, 15% saving money for childbirth, 9.6% identifying skilled birth attendant and only 5.3% had identified transportation for delivery [32].

Regarding the factors influencing BPCR, this study found that ANC checkups, place of birth, and awareness of serious warning signs during pregnancy and postpartum had a significantly positive association with BPCR. Place of birth showed a positive association with BPCR, those mothers who had given birth at home were twice likely to be not prepared for birth and its complications than those who had given birth at a health institution. This is nearly similar to a study conducted in Cameroon, Central Africa. This could be due to similar socio-cultural barriers and lack of decision power regarding birth or a lack of information to seek care in a health facility [37].

There was a statistically significant relationship between BPCR awareness of serious warning signs during pregnancy and postpartum period with those recently delivered mothers who were not knowledgeable about serious warning signs during pregnancy being only half prepared for birth and its complications as compared to those who were knowledgeable. This is lower than studies done in Tanzania, which highlights that the less knowledge on BPCR they have, the less they practice it indicating that creating awareness of serious warning signs during pregnancy and childbirth is very important for good BPCR [38].

The number of ANC visits improves BPCR awareness, and there was a significant association between BPCR and the number of ANC checkups. Those women who had ≥ 4 ANC visits were nearly four times more likely to be well prepared for birth and its complications, compared to women who had less than four ANC visits.

## Recommendations

The proportion of women who were well prepared for birth and its complications were found to be low in rural areas, but having frequent ANC visits improved BPCR. It is recommended to organize community-based education and improve the quality of MNCH services which will improve BPCR among women in rural Sindh. It is also recommended that community-based basic obstetric care such as ANC visits should be incorporated in the policymaking by involving the local management in the decision-making. Empowering women through increasing educational training is an important step in enhancing BPCR and reducing delays in obtaining skilled obstetric care. Further, we need to target rural populations where awareness is lacking regarding BPCR and implement designed health education programs to improve the status of BPCR and the conditions under which they should be used.

## Acknowledgments

I gratefully acknowledge the valuable support provided by APPNA Institute of Public Health and the JHPIEGO, Karachi, to carry out a study at Thatta. My sincere appreciation and thanks to all the supportive staff of JHPIEGO Thatta, especially Miss. Ambreen Gul and Aijaz Jhakro for their help and support during my field work and thanks for the support of all the respondents, for without them this research would not have been possible.

## Author Contributions

**Conceptualization:** Ruquia Noor, Farhana Shahid.

**Data curation:** Ruquia Noor, Farhana Shahid, Muhammad Imran.

**Formal analysis:** Ruquia Noor, Farhana Shahid, Muhammad Zafar Iqbal Hydrie.

**Investigation:** Farhana Shahid, Muhammad Imran.

**Methodology:** Ruquia Noor, Farhana Shahid, Muhammad Imran, Syed Hassan Bin Usman Shah.

**Project administration:** Ruquia Noor.

**Supervision:** Farhana Shahid, Muhammad Zafar Iqbal Hydrie, Muhammad Imran.

**Validation:** Ruquia Noor, Farhana Shahid, Syed Hassan Bin Usman Shah.

**Visualization:** Ruquia Noor, Farhana Shahid.

**Writing – original draft:** Ruquia Noor, Muhammad Zafar Iqbal Hydrie, Muhammad Imran.

**Writing – review & editing:** Muhammad Zafar Iqbal Hydrie, Muhammad Imran, Syed Hassan Bin Usman Shah.

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
