## [Decision Letter · Decision Letter 0]

30 Oct 2021

PONE-D-21-20638Factors influencing Birth preparedness and Complication Readiness among child bearing age women in Thatta district, SindhPLOS ONE

Dear Dr. Hydrie,

Thank you for submitting your manuscript to PLOS ONE. After careful consideration, we feel that it has merit but does not fully meet PLOS ONE’s publication criteria as it currently stands. Therefore, we invite you to submit a revised version of the manuscript that addresses the points raised during the review process.

Kindly using recent literature explore the situation of birth preparedness and maternal health situation of Sindh province. Provide a flowchart to show the sample selection process and mention whether you performed any normality tests.

We look forward to receiving your revised manuscript.

Kind regards,

Russell Kabir, PhD

Academic Editor

PLOS ONE

Journal Requirements:

Reviewers' comments:

Reviewer's Responses to Questions

**Comments to the Author**

1. Is the manuscript technically sound, and do the data support the conclusions?

Reviewer #1: Yes

Reviewer #2: Partly

Reviewer #3: Yes

2. Has the statistical analysis been performed appropriately and rigorously? 

Reviewer #1: Yes

Reviewer #2: No

Reviewer #3: Yes

3. Have the authors made all data underlying the findings in their manuscript fully available?

Reviewer #1: Yes

Reviewer #2: Yes

Reviewer #3: No

4. Is the manuscript presented in an intelligible fashion and written in standard English?

Reviewer #1: Yes

Reviewer #2: No

Reviewer #3: Yes

5. Review Comments to the Author

Reviewer #1: 1. No major issue was identified. Rationale for conducting study is nicely explained.

2. Methodology- A mention of what constitutes a cluster unit, cluster size and number of clusters in multi-stage cluster sampling would have been better.

3. Introduction and discussion can be made shorter & crisper by avoiding reputations of similar arguments.

Reviewer #2: 1. The data and results could have been represented in a better way with all the values.

2. There is no reference for the information stated in the sentence "Maternal deaths are more prevalent in developing countries due to lack of BPCR" in the introduction. Please add a reference.

3. No exact criteria is defined for the inclusion or the exclusion of the subjects in the paper. For the selection of the subjects,

it's stated "multistage cluster sampling" was used in the methodology. However, why the exact numbers are not

specified for each stage of cluster sampling?

3. There is a typo in the sentence "The practice of birth preparedness and complication readiness in this study was found to be low i.e. 21.2%, as compared to other developing countries tudies such as Ethiopia (37%), India (58.5%) and in West Bengal" in Discussion. Wrong information: West Bengal is not a country, it's a state in the country, India. Please correct the typo and the information.

4. Please correct the typo in the sentence "This variation might be due to socio-cultural and study settings differences as well as d implementation of related health programs".

5. It will be good to add some recent reviews or references of BPCR (from2021) in the paper.

Reviewer #3: The article addresses one of the crucial issues faced by women in the society, especially in the developing world. The authors have used appropriate methodology and statistical analytical techniques to examine the issue. Some justification on sampling the women who gave birth to children in past 12 months would have enhanced the value of research.

6. PLOS authors have the option to publish the peer review history of their article (what does this mean?). If published, this will include your full peer review and any attached files.

Reviewer #1: **Yes: **Rakesh Kumar

Reviewer #2: No

Reviewer #3: **Yes: **Kiran Pandya

---

## [Author Response · Author response to Decision Letter 0]

26 Nov 2021

Following are the answers of the raised points of academic editor and reviewers.

1. Kindly using recent literature explore the situation of birth preparedness and maternal health situation of Sindh province

The recent literature of birth preparedness (Reference number 12, 19, 20) and maternal health situation of Sindh (Reference number 15, 17, 18) have added.

2. Provide a flowchart to show the sample selection process

The flow chart of multistage cluster sampling have added

3. Mention whether you performed any normality tests.

The normality test by mean median mode were performed and data was normally distributed

4. Please review your reference list to ensure that it is complete and correct.

All references have completed and corrected

Specific Reviewers Response from the Authors

Reviewer #1: 

1. No major issue was identified. Rationale for conducting study is nicely explained.

Ans: Thank you. 

2. Methodology- A mention of what constitutes a cluster unit, cluster size and number of clusters in multi-stage cluster sampling would have been better.

Ans: Flow chart of cluster sampling have added for better understanding.

3. Introduction and discussion can be made shorter & crisper by avoiding reputations of similar arguments.

Ans: The reputations of similar argument have removed

Reviewer #2: 

1. The data and results could have been represented in a better way with all the values.

Ans: The data and all the results have represented with all values

2. There is no reference for the information stated in the sentence "Maternal deaths are more prevalent in developing countries due to lack of BPCR" in the introduction. Please add a reference.

Ans: The reference number 10 have added

3. No exact criteria is defined for the inclusion or the exclusion of the subjects in the paper. For the selection of the subjects,it's stated "multistage cluster sampling" was used in the methodology. However, why the exact numbers are not specified for each stage of cluster sampling?

Ans: Those women who had delivered in the last 12 months were included in this study due to recall bias, and multistage cluster sampling were used which is further clearly defined by added flow chart of cluster sampling, and exact number of 65 women were added in the third stage of sampling by random sampling 

3. There is a typo in the sentence "The practice of birth preparedness and complication readiness in this study was found to be low i.e. 21.2%, as compared to other developing countries tudies such as Ethiopia (37%), India (58.5%) and in West Bengal" in Discussion. Wrong information: West Bengal is not a country, it's a state in the country, India. Please correct the typo and the information.

Ans: The typo mistake and information has been corrected.

4. Please correct the typo in the sentence "This variation might be due to socio-cultural and study settings differences as well as d implementation of related health programs".

Ans: The typo mistake has been corrected.

5. It will be good to add some recent reviews or references of BPCR (from2021) in the paper.

Ans: Recent reviews or references of BPCR (2021) no: (12,19,22,23,24,28,31,35 and 36)

Reviewer #3: 

The article addresses one of the crucial issues faced by women in the society, especially in the developing world. The authors have used appropriate methodology and statistical analytical techniques to examine the issue. Some justification on sampling the women who gave birth to children in past 12 months would have enhanced the value of research.

Ans: Due to recall bias those women who gave birth in past 12 months were included in this study, it is very difficult for women to recall previous complications related to pregnancy and delivery.

---

## [Decision Letter · Decision Letter 1]

4 Aug 2022

PONE-D-21-20638R1Factors influencing Birth preparedness and Complication Readiness among child bearing age women in Thatta district, SindhPLOS ONE

Dear Dr. Hydrie,

Thank you for submitting your manuscript to PLOS ONE. After careful consideration, we feel that it has merit but does not fully meet PLOS ONE’s publication criteria as it currently stands. Therefore, we invite you to submit a revised version of the manuscript that addresses the points raised during the review process.

Although the reviewers are happy with the revision, I have concerns about the missing details on the data collection procedures. Specifically, please report details of A) all the variables measured as part of the study B) how they were measured. This information should be provided in full for reproducibility purpose. In addition, 3D effects in plots can bias and hinder interpretation of values, so avoid them in cases where regular plots are sufficient to display the data. We suggest replacing the plots in your manuscript with 2d versions.

We look forward to receiving your revised manuscript.

Kind regards,

Jianhong Zhou

Staff Editor

PLOS ONE

Journal Requirements:

Please review your reference list to ensure that it is complete and correct. If you have cited papers that have been retracted, please include the rationale for doing so in the manuscript text, or remove these references and replace them with relevant current references. Any changes to the reference list should be mentioned in the rebuttal letter that accompanies your revised manuscript. If you need to cite a retracted article, indicate the article’s retracted status in the References list and also include a citation and full reference for the retraction notice.:

Reviewers' comments:

Reviewer's Responses to Questions

**Comments to the Author**

1. If the authors have adequately addressed your comments raised in a previous round of review and you feel that this manuscript is now acceptable for publication, you may indicate that here to bypass the “Comments to the Author” section, enter your conflict of interest statement in the “Confidential to Editor” section, and submit your "Accept" recommendation.

Reviewer #2: All comments have been addressed

Reviewer #3: All comments have been addressed

2. Is the manuscript technically sound, and do the data support the conclusions?

Reviewer #2: Yes

Reviewer #3: Yes

3. Has the statistical analysis been performed appropriately and rigorously? 

Reviewer #2: Yes

Reviewer #3: Yes

4. Have the authors made all data underlying the findings in their manuscript fully available?

Reviewer #2: Yes

Reviewer #3: No

5. Is the manuscript presented in an intelligible fashion and written in standard English?

Reviewer #2: Yes

Reviewer #3: Yes

6. Review Comments to the Author

Reviewer #2: (No Response)

Reviewer #3: (No Response)

7. PLOS authors have the option to publish the peer review history of their article (what does this mean?). If published, this will include your full peer review and any attached files.

Reviewer #2: **Yes: **Deblina Patra Bhattacharya

Reviewer #3: **Yes: **Kiran Pandya

---

## [Author Response · Author response to Decision Letter 1]

29 Aug 2022

Editor/Reviewer Comments:

We thank and welcome the comments made by the editor and reviewers. We believe these comments will greatly improve this manuscript. Please see below for detailed responses to each comment. We have revised our manuscript as per their comments and suggestions.

Editors Comments

Comment 1:

Missing details on the data collection procedures. Specifically, please report details of A) all the variables measured as part of the study B) how they were measured. 

Response:

We agree with the editor's suggestion. We have edited the methodology to include details of all the variables and how they were calculated (clean draft; Methodology, page 7, paragraph 2).

Comment 2:

In addition, 3D effects in plots can bias and hinder interpretation of values, so avoid them in cases where regular plots are sufficient to display the data. We suggest replacing the plots in your manuscript with 2d versions. 

Response:

We agree with the editor's suggestion of changing the graphs to 2D. We have revised figures 2 and 3 on pages 10 and 13.

---

## [Editor Report · Decision Letter 2]

13 Sep 2022

Factors influencing Birth preparedness and Complication Readiness among child bearing age women in Thatta district, Sindh

PONE-D-21-20638R2

Dear Dr. Hydrie,

We’re pleased to inform you that your manuscript has been judged scientifically suitable for publication and will be formally accepted for publication once it meets all outstanding technical requirements.

Kind regards,

Jianhong Zhou

Staff Editor

PLOS ONE
---

## [Editor Report · Acceptance letter]

21 Sep 2022

PONE-D-21-20638R2 

Factors influencing Birth preparedness and Complication Readiness among childbearing age women in Thatta district, Sindh 

Dear Dr. Hydrie:

I'm pleased to inform you that your manuscript has been deemed suitable for publication in PLOS ONE. Congratulations! Your manuscript is now with our production department. 

Kind regards, 

on behalf of

Jianhong Zhou 

Staff Editor

PLOS ONE